# Profiling of Dust and Urban Haze Mass Concentrations during the 2019 National Day Parade in Beijing by Polarization Raman Lidar

Zhuang Wang [1,2], Cheng Liu [1,2,3,4,5], Yunsheng Dong [1], Qihou Hu [1], Ting Liu [6], Yizhi Zhu [1,2] and Chengzhi Xing [1,*]

1 Key Lab of Environmental Optics and Technology, Anhui Institute of Optics and Fine Mechanics, Hefei Institutes of Physical Science, Chinese Academy of Sciences, Hefei 230031, China; zhwang95@mail.ustc.edu.cn (Z.W.); chliu81@ustc.edu.cn (C.L.); ysdong@aiofm.ac.cn (Y.D.); qhhu@aiofm.ac.cn (Q.H.); yzz2017@mail.ustc.edu.cn (Y.Z.)
2 Department of Precision Machinery and Precision Instrumentation, University of Science and Technology of China, Hefei 230026, China
3 Center for Excellence in Regional Atmospheric Environment, Institute of Urban Environment, Chinese Academy of Sciences, Xiamen 361021, China
4 Key Laboratory of Precision Scientific Instrumentation of Anhui Higher Education Institutes, University of Science and Technology of China, Hefei 230026, China
5 Anhui Province Key Laboratory of Polar Environment and Global Change, University of Science and Technology of China, Hefei 230026, China
6 Department of Environmental Science and Engineering, University of Science and Technology of China, Hefei 230026, China; tliu95@mail.ustc.edu.cn
* Correspondence: xingcz00@mail.ustc.edu.cn

**Abstract:** The polarization–Raman Lidar combined sun photometer is a powerful method for separating dust and urban haze backscatter, extinction, and mass concentrations. The observation was performed in Beijing during the 2019 National Day parade, the particle depolarization ratio at 532 nm and Lidar ratio at 355 nm are $0.13 \pm 0.05$ and $52 \pm 9$ sr, respectively. It is the typical value of a mixture of dust and urban haze. Here we quantify the contributions of cross-regional transported natural dust and urban haze mass concentrations to Beijing's air quality. There is a significant correlation between urban haze mass concentrations and surface $PM_{2.5}$ ($R = 0.74$, $p < 0.01$). The contributions of local emissions to air pollution during the 2019 National Day parade were insignificant, mainly affected by regional transport, including urban haze in North China plain and Guanzhong Plain (Hebei, Tianjin, Shandong, and Shanxi), and dust aerosol in Mongolia regions and Xinjiang. Moreover, the trans-regional transmission of natural dust dominated the air pollution during the 2019 National Day parade, with a relative contribution to particulate matter mass concentrations exceeding 74% below 4 km. Our results highlight that controlling anthropogenic emissions over regional scales and focusing on the effects of natural dust is crucial and effective to improve Beijing's air quality.

**Keywords:** Raman Lidar; mass concentrations; dust; urban haze; depolarization ratio; MODIS

## 1. Introduction

Mineral (desert) dust and urban haze are the main components of the atmospheric aerosol system over the North China Plain (NCP). NCP is one of the areas with the most anthropogenic aerosol pollution in China [1,2]. Asian dust originates in the interior of Eurasia (Mongolia and the Taklamakan Desert) and travels eastward [3]. Large amounts of mineral dust can be transported from these deserts to the NCP over long distances in the lower free troposphere within a few days. The turbulent exchange process at the interface of the free troposphere and the planetary boundary layer results in the effective mixing of dust with urban haze, presenting vertical stratification of aerosols [4]. Thus, it is essential to quantify the impact of these aerosols on air quality in Beijing.

Polarization–Raman–Lidar (PRL) allows the identification of depolarized coarse mode particles (non–spherical particles), such as volcanic ash and desert dust, as well as non–depolarized fine mode particles (spherical particles), such as urban haze [5–12]. Assuming that aerosol particles are mixed externally, this technique further allows us to estimate the contribution of dust and spherical aerosols to the backscatter coefficient. With a suitable lidar ratio (also called extinction to backscatter ratio), the extinction coefficient of dust and spherical aerosol can be obtained [7,8,13,14]. This method is simple, reliable, and can maintain high-quality measurements for a long time. In addition, the sun photometer can provide microscopic physical properties of aerosol particles, such as aerosol optical depth (AOD) related to the fine and coarse modes, volume concentrations of fine and coarse mode particles [15,16]. These measurements provide a conversion factor from extinction coefficient to mass concentrations [17–21]. Therefore, combined with sun photometer observations, we can retrieve coarse particles and fine particles' mass concentration profiles from the extinction profiles.

As the capital of China, Beijing is a typical rapidly developing megalopolis. The north and west sides are respectively surrounded by Yanshan and Taihang Mountain, effectively intercepting air pollutants [22–24]. A large amount of local emissions and the transportation of regional air pollutants have continuously deteriorated Beijing's air quality under unfavorable weather conditions [25]. Thus, Beijing has implemented strict emission control efforts to improve air quality when hosting major events, such as the 2008 Olympic Games [26,27] and the 2014 Asia–Pacific Economic Cooperation [28]. Previous studies have focused on urban haze, especially the change of anthropogenic air pollutants in the early, middle, and late stages of major events held in Beijing, and focused on the reduction of various aerosol components (sulfate, nitrate, organic components, etc.) during major events. This paper focuses on the contribution of cross-regional transmission of dust and urban haze to Beijing's air quality during the 2019 National Day military parade. The air pollution during the 2019 National Day military parade was different from former studies. There was serious air pollution on the day of the 2019 National Day military parade. Firstly, the contribution of natural sources increased significantly, mainly affected by the dust transmitted from the northwest, and secondly, there was the transmission of urban haze in the south. In addition, there were no large–scale emission restrictions during the 2019 National Day military parade, and only emissions in Beijing and adjacent areas were controlled, such as traffic control and stopping all construction activities in the urban area (http://www.kanxsw.com/cheguan/shenzhenditie_35689.html, in Chinese, access on: 23 August 2021). This provides a good opportunity to investigate the impact of regional transmission of natural dust and urban haze on Beijing's air quality.

In this study, the PRL was used to determine the particle depolarization ratio at 532 nm ($PLDR_{532}$) and backscatter coefficient at 532 nm ($Bac_{532}$) before and after the 2019 National Day military parade (13 September to 9 October 2019). Combining the surface fine particulate matter ($PM_{2.5}$) and urban haze mass concentrations retrieved by PRL, the urban haze mass concentrations measured by PRL from 13 September to 9 October 2019 were tested and compared. Based on the daily average urban haze and dust mass concentration profiles during the 2019 National Day military parade, the relative contribution of regional transported natural dust and urban haze mass concentrations to Beijing's air pollution was quantitatively analyzed. Finally, the spatial correlation analysis of AOD measured by MODIS was performed to investigate the potential sources of air pollution in Beijing. The results enhance the understanding of the formation and growth of air pollution in Beijing and provide a scientific basis for Beijing's air pollution control and management.

## 2. Instruments and Materials

### 2.1. Polarization–Raman Lidar

In this study, the zenith-pointed ground-based PRL was performed as a primarily remote sensing instrument. The measurement site is located at the Beijing Meteorological Observation Center (BMOC, 39.80°N, 116.47°E), near the Fifth south ring of Beijing, with

dense traffic and large vehicle flow. The measurement campaign was conducted from 13 September to 9 October 2019 to observe urban haze and the regional transport of Asian dust. During our observation period, there was light rain on the night of 3 October, and the rest was sunny.

The PRL light source produces two wavelengths: 355 nm and 532 nm. PRL is equipped with a nitrogen (387 nm) Raman scattering channel for measuring the night–time lidar ratio at 355 nm (LR$_{355}$), 19:00–6:00 local time (LT) [29,30]. The backscatter signal at 532 nm is divided into cross and parallel polarization components for distinguishing urban haze and Asian dust. The frequency of the PRL data collector is 20 M HZ, the vertical spatial resolution is 7.5 m, and the time resolution is set to 15 min. In addition, PRL adopts an off-axis structure, and the complete overlap of the transmitting system and receiving system starts about 250 m above ground. Here, we used the night–time average backscatter signal to retrieve the lidar ratio due to the low signal-to-noise ratio of inelastic backscatter signals. More details about the PRL system and data retrieval can be found in our previous studies [4,31].

### 2.2. MODIS

The Moderate Resolution Imaging Spectrometer (MODIS) boarded on the Aqua satellite can provide global AOD. This study used MCD19A2 data (https://lpdaac.usgs.gov/products/mcd19a2v006/, access on: 23 August 2021) with a spatial resolution of $1 \times 1$ km [32,33]. MCD19A2 product is a MODIS Terra and Aqua combined Multi-angle Implementation of Atmospheric Correction (MAIAC) Land AOD gridded Level 2 product. The green band AOD at 550 nm from 24 September to 3 October was used to evaluate the air pollution in Beijing and surrounding regions.

### 2.3. CALIPSO

The Cloud-Aerosol Lidar and Infrared Pathfinder Satellite Observations (CALIPSO) lidar was launched in 2006, and it is the first active remote sensing instrument to observe the vertical profile of aerosol and clouds [34,35]. The satellite's orbit passes over Beijing at about 2:20 and 13:00, two CALIPSO overpasses were found on 1 October 2019. In this study, level 2 aerosol products ("*CAL_LID_L2_05kmAPro–Standard–V4–20.*"), i.e., extinction coefficient at 532 nm and PLDR$_{532}$ were used. In addition, CALIOP Level–2 vertical feature mask (VFM) product ("*CAL_LID_L2_VFM–Standard–V4–20.*") was also presented to identify the Asian dust layers and urban haze over North China plain.

### 2.4. WRF–Chem

The meteorology parameters (wind speed and direction) from 23 September to 6 October 2019 were simulated by Weather Research and Forecasting (WRF) model coupled with Chemistry (WRF–Chem). There are 20 layers below 3 km to well describe the meteorology parameters vertical structure. Details of model configuration and data validation can be found in our previous studies [4,36]. The comparison of wind speed (u and v components) between WRF–Chem simulations and observations can be found in Supplementary Materials (Figures S1 and S2 in the Supplementary).

### 2.5. Surface Air Pollutants Concentrations

The hourly particle matter mass concentrations (PM$_{2.5}$ and PM$_{10}$) and gaseous pollutants (SO$_2$, NO$_2$, CO, and O$_3$) from 13 September to 9 October 2019 are collected from Beijing Municipal Ecological and Environmental Monitoring Center (http://www.bjmemc.com.cn/, access on: 23 August 2021). We chose an official environmental monitoring station closest to BMOC, which is located about 15 km north of BMOC (station number 1005A, 39.97°N, 116.47°E).

### 2.6. Backward Trajectory

We use the Hybrid Single-Particle Lagrangian Integrated Trajectory (HYSPLIT) model of the National Center for Environmental Prediction (NCEP) GDAS data product to calculate the 96–hour air mass backward trajectories [37] and explore the possible sources of air pollutants on 1 October 2019. The initial heights of each air mass backward trajectory are 500 m, 1500 m, 2500 m, and 3500 m.

## 3. Methodology

### 3.1. Retrieval of the Mass Concentration of Dust and Urban Haze by PRL

There are two steps to obtain the mass concentrations of dust and urban haze by PRL. In the first step, separate the dust and urban haze contributions to the total backscatter coefficient [7,8,13,14]. The key principle of this step is to obtain prior information on the PLDR of the dust and urban haze. In the second step, convert the dust and urban haze backscatter coefficient to mass concentrations [17,21].

The backscatter coefficient $\beta$ can be obtained from the Fernald method [38]. In order to obtain a more accurate backscatter coefficient, the range-dependent $LR_{355}$ measured at night was used to calculate the $Bac_{532}$. The volume linear depolarization ratio $\delta_v$ (VLDR) is retrieved from the ratio of the cross-polarized backscatter coefficient $\beta^{\perp}$ to the parallel polarized backscatter coefficient $\beta^{\parallel}$ [9].

$$\delta_v = \beta^{\perp}/\beta^{\parallel} \tag{1}$$

First of all, the backscatter coefficient and VLDR are contributed by atmospheric particles and molecules. We can separate the contribution of particles and molecules to the backscatter coefficient and VLDR.

$$\beta^{\perp} = \beta_p^{\perp} + \beta_m^{\perp}, \qquad \beta^{\parallel} = \beta_p^{\parallel} + \beta_m^{\parallel} \tag{2}$$

$$\beta_p = \beta_p^{\perp} + \beta_p^{\parallel}, \qquad \beta_m = \beta_m^{\perp} + \beta_m^{\parallel} \tag{3}$$

$$\delta_p = \beta_p^{\perp}/\beta_p^{\parallel}, \qquad \delta_m = \beta_m^{\perp}/\beta_m^{\parallel} \tag{4}$$

The subscript $p$ and m represent the atmospheric particles and molecules, respectively. Combine Equations (3) and (4), the following relationships can be obtained.

$$\beta_p^{\parallel} = \frac{\beta_p}{1+\delta_p}, \qquad \beta_m^{\parallel} = \frac{\beta_m}{1+\delta_m} \tag{5}$$

$$\beta_p^{\perp} = \frac{\beta_p \delta_p}{1+\delta_p}, \qquad \beta_m^{\perp} = \frac{\beta_m \delta_m}{1+\delta_m} \tag{6}$$

We replace the $\beta^{\perp}$ and $\beta^{\parallel}$ in Equation (1) by respective Equations (2), (5), and (6) and obtain.

$$\delta_v = \frac{\frac{\beta_p \delta_p}{1+\delta_p} + \frac{\beta_m \delta_m}{1+\delta_m}}{\frac{\beta_p}{1+\delta_p} + \frac{\beta_m}{1+\delta_m}} \tag{7}$$

Simple the Equation (7)

$$\delta_v = \frac{\beta_m \delta_m(1+\delta_p) + \beta_p \delta_p(1+\delta_m)}{\beta_m(1+\delta_p) + \beta_p(1+\delta_m)} \tag{8}$$

After rearranging the Equation (8), the PLDR was obtained

$$\delta_p = \frac{\beta_m(\delta_v - \delta_m) + \beta_p \delta_v(1+\delta_m)}{\beta_m(\delta_m - \delta_v) + \beta_p(1+\delta_m)} \tag{9}$$

The atmospheric molecule depolarization ratio $\beta_m$ can be set to 0.014 according to Murayama et al. 1999 [39]. In the same way, if the atmospheric particulate matter is mixed by two aerosols, such as dust aerosol and urban haze, we can separate the contribution of dust aerosol and urban haze to the backscatter coefficient and PLDR. The PLDR is defined as the ratio of the cross-polarized atmospheric particles backscatter coefficient $\beta_p^{\perp}$ to the parallel polarized atmospheric particles backscatter coefficient $\beta_p^{\parallel}$ [9].

$$\delta_p = \beta_p^{\perp} / \beta_p^{\parallel} \tag{10}$$

The atmospheric particles backscatter coefficient and PLDR are contributed by urban haze and dust aerosol. Thus,

$$\beta_p^{\perp} = \beta_u^{\perp} + \beta_d^{\perp}, \qquad \beta_p^{\parallel} = \beta_u^{\parallel} + \beta_d^{\parallel} \tag{11}$$

$$\beta_d = \beta_d^{\perp} + \beta_d^{\parallel}, \qquad \beta_u = \beta_u^{\perp} + \beta_u^{\parallel} \tag{12}$$

$$\delta_d = \beta_d^{\perp} / \beta_d^{\parallel}, \qquad \delta_u = \beta_u^{\perp} / \beta_u^{\parallel} \tag{13}$$

The subscript u and d represent the urban haze and dust aerosol, respectively. Combine Equations (12) and (13), the following relationships can be obtained.

$$\beta_u^{\parallel} = \frac{\beta_u}{1 + \delta_u}, \qquad \beta_u^{\parallel} = \frac{\beta_u}{1 + \delta_u} \tag{14}$$

$$\beta_d^{\perp} = \frac{\beta_d \delta_d}{1 + \delta_d}, \qquad \beta_d^{\perp} = \frac{\beta_d \delta_d}{1 + \delta_d} \tag{15}$$

We replace the $\beta_p^{\perp}$ and $\beta_p^{\parallel}$ in Equation (10) by respective Equations (11), (14), and (15) and obtain.

$$\delta_p = \frac{\beta_u \delta_u (1 + \delta_d) + \beta_d \delta_d (1 + \delta_u)}{\beta_u (1 + \delta_d) + \beta_d (1 + \delta_u)} \tag{16}$$

Replace $\beta_u$ with $\beta_p - \beta_d$ in Equation (16), and rearranging the equation, the backscatter coefficient of dust can be obtained

$$\beta_d = \beta_p \frac{(\delta_p - \delta_u)(1 + \delta_d)}{(\delta_d - \delta_u)(1 + \delta_p)} \tag{17}$$

In Equation (17), to obtain the dust backscatter coefficient $\beta_d$, the particle backscatter coefficient $\beta_p$ and PLDR $\delta_p$ need to be calculated first. Then, the dust depolarization ratio $\delta_d$ and urban haze depolarization ratio $\delta_u$ need to be estimated. According to our previous measurements, the depolarization ratio of dust aerosol and urban haze was assigned to 0.322 and 0.063, respectively [31]. It is noteworthy that, if $\delta_p < \delta_u$, we set $\beta_p = 0$, and if $\delta_p > \delta_d$, we set $\beta_d = \beta_p$. Moreover, the dust extinction coefficient $\sigma_d$ can be estimated from the multiply of dust backscatter coefficient and dust lidar ratio $S_d$.

$$\sigma_d = S_d \times \beta_d \tag{18}$$

Secondly, the mass concentrations of dust aerosol $m_d$ and the mass concentrations of urban haze $m_u$ are derived combine sun photometer observations.

$$m_d = \rho_d \times \overline{(v_d / \tau_d)} \times \sigma_d \tag{19}$$

$$m_u = \rho_u \times \overline{(v_u / \tau_u)} \times \sigma_u \tag{20}$$

The $\rho_d$ and $\rho_u$ are particle density of dust aerosol and urban haze, respectively. The terms $v_u$ and $v_d$ denote the fine-mode and coarse-mode volume concentrations measured by sun photometer, respectively. $\tau_u$ and $\tau_d$ are AOD for fine-mode and coarse-mode particles

observed by Sun photometer, respectively. $\sigma_u$ is the urban haze lidar ratio. According to the sun photometer observations at the same site, the extinction-to-volume conversion factors $\overline{v_d/\tau_d}$ and $\overline{v_u/\tau_u}$ are set to 1.1 μm and 0.14 μm, respectively [40]. The detailed input parameters for mass concentration retrieval are shown in Table 1.

**Table 1.** Input parameters in dust and urban haze mass concentrations retrieval.

| Parameter | Value | Reference |
|---|---|---|
| Urban haze lidar ratio $S_u$ | $55.2 \pm 10.4$ sr | [31] |
| Asian dust lidar ratio $S_d$ | $43.0 \pm 5.2$ sr | [31] |
| Urban haze depolarization ratio $\delta_u$ | $0.063 \pm 0.022$ | [31] |
| Asian dust depolarization ratio $\delta_d$ | $0.322 \pm 0.055$ | [31] |
| Urban haze mass density $\rho_u$ | $1.5 \pm 0.3$ g/cm$^3$ | [17] |
| Asian dust mass density $\rho_d$ | $2.6 \pm 0.6$ g/cm$^3$ | [17] |
| Urban haze conversion factor $\overline{v_u/\tau_u}$ | $0.14 \pm 0.02$ μm | [40] |
| Asian dust conversion factor $\overline{v_d/\tau_d}$ | $1.1 \pm 0.22$ μm | [40] |

*3.2. Retrieval Uncertainties of the Mass Concentration of Dust and Urban Haze*

Uncertainties in the retrieve of dust and urban haze mass concentrations mainly include two parts: (a) uncertainty of PRL observations, i.e., PLDR$_{532}$ and Bac$_{532}$, it primarily depends on the signal-to-noise ratio of the PRL backscatter signal [41]. The uncertainty of the Bac$_{532}$ and PLDR$_{532}$ below 3 km during our measurements is about 2–15% and 2–20%, respectively. (b) The input parameters are listed in Table 1, i.e., the assumption of dust and urban haze PLDR$_{532}$ threshold. Uncertainties in retrieved mass concentrations are calculated by the law of error propagation. Each input parameter in Table 1 changes within the specified value range (here we set one standard deviation), while other input parameters are set to a fixed value, and the sensitivity of the parameters in Table 1 are tested. Generally, the uncertainty of mass concentrations below 3 km ranges from 15% to 70%.

It is also necessary to discuss the water-uptake effects. The uncertainties caused by the water-uptake effect primarily contain three parts. Firstly, the urban haze and Asian dust mass density shown in Table 1 vary with relative humidity and depend on the actual composition of the aerosol mixture. Secondly, urban haze and Asian dust conversion factors shown in Table 1 may increase due to the elevated relative humidity [17]. Finally, the shape characteristics of the originally non-spherical particles change with the increase in water content. The basis for our separation is a significant difference in the shape of the two mixed aerosol particles. Therefore, the mass concentrations of urban haze and dust particles separated by this method may not be trustworthy under high relative humidity (>80%). The high relative humidity must be kept in mind when retrieving the mass concentrations mixed by two kinds of aerosol types. Overall, the uncertainties caused by water-uptake effects in the retrieval of dry but hygroscopic urban haze were estimated to be <25%. Dust aerosol is almost hydrophobic, the water uptake does not play a role here [17,18]. Fortunately, the relative humidity is usually less than 75% from 23 September to 3 October (Figure S3 in the Supplementary).

Figure 1 presents a case study of Bac$_{532}$ and PLDR$_{532}$ profiles measured at 0:01 on 1 October 2019 (LT). The high-loading aerosols are concentrated below 2.3 km, and the PLDR$_{532}$ is between 0.10 and 0.21, which are mixed particles of dust and urban haze. Uncertainties in the retrieved Bac$_{532}$ and PLDR$_{532}$ profiles under 2.3 km are 1–15% and 1–16%, respectively. After the Bac$_{532}$ separation and successive conversion of Bac$_{532}$ to mass concentrations, the relative error reaches 18–56%.

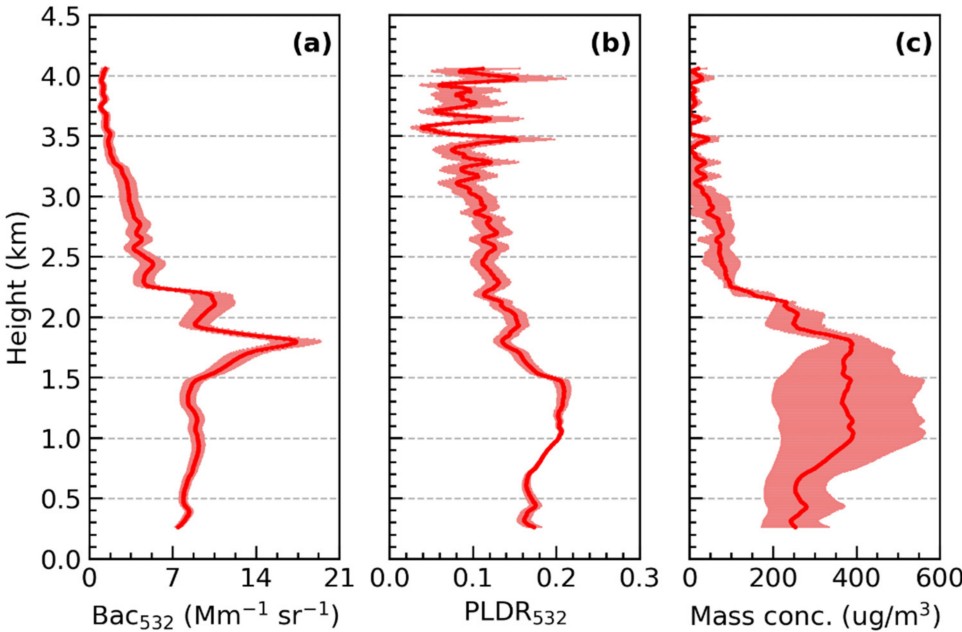

**Figure 1.** PRL observed (**a**) backscatter coefficient at 532 nm (Bac$_{532}$), (**b**) PLDR at 532 nm (PLDR$_{532}$), (**c**) dust mass concentrations at 0:01 on 1 October 2019 (LT). PRL observed profile is the solid red line and the envelope represents the errors at each altitude. Error bars are calculated from the law of error propagation, which primarily depends on the signal-to-noise ratio of the PRL backscatter signal and input parameters given in Table 1.

### 3.3. Spatial Correlation Analysis of MODIS Aerosol Optical Depth

To evaluate the impact of regional transport on air quality in Beijing. The MODIS daily AOD products with spatial resolution $1° \times 1°$ from 24 September to 3 October are used to investigate the aerosol loading in Beijing and influential sources. A spatial correlation analysis method based on AOD is presented. Based on the daily AOD of the observation station, the Pearson correlation analysis was carried out between the AOD time series of each investigated pixel and the AOD time series of the BMOC.

$$R[i, j] = \text{Corre}_l(p_{\text{site}}, p_{[i,j]}) \qquad (21)$$

[i, j] represents the investigated pixel in row i and column j. P$_{\text{site}}$ represents the AOD time series at BMOC. P$_{[i, j]}$ represents the time series of AOD on investigated pixel [i, j]. *Corre* denotes the Pearson correlation functions, and the subscript *l* represents the lagging days. R[i, j] represents the AOD correlation coefficient between BMOC and investigated pixel [i, j] in a certain research period. To avoid outliers due to missing data, for example, two points are used to calculate the Pearson correlation, we set a threshold value of more than 5 for valid satellite observations for correlation analysis.

## 4. Results

### 4.1. Overview of Aerosol Vertical Distribution

The vertical distribution of Bac$_{532}$ and PLDR$_{532}$ during the whole observation period is shown in Figure 2. Four haze episodes were captured according to the high Bac$_{532}$ (>5 Mm$^{-1}$ sr$^{-1}$) in the lower lidar layer, i.e., 15 September to 18 September, 19 September to 23 September, 24 September to 3 October, and 5 October to 8 October. Surface PM$_{10}$ and PM$_{2.5}$ mass concentrations also present the same period air pollution cycles. Moreover, there is a significant positive correlation between surface PM$_{10}$ and PM$_{2.5}$ mass concentrations (Pearson correlation R = 0.84, *p* < 0.01), which are typical saw–tooth cycles [42], indicating the haze episodes are usually controlled by meteorological parameters, such as cross-regional transmission of air pollutants.

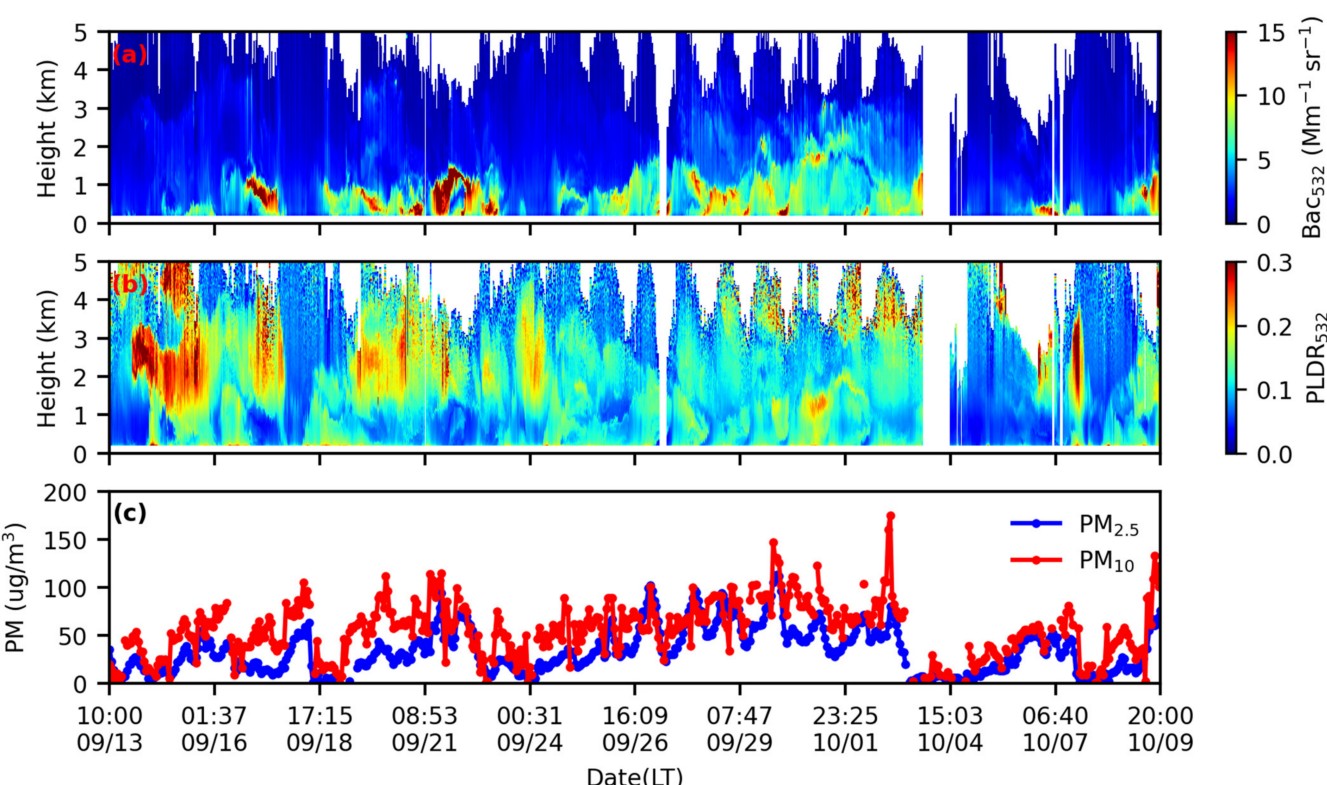

**Figure 2.** Period air pollution cycles in Beijing from 13 September to 9 October 2019. Time–height plots of the (**a**) $Bac_{532}$ and (**b**) $PLDR_{532}$ are measured by PRL. (**c**) Time series of surface $PM_{2.5}$ and $PM_{10}$ mass concentrations.

The $PLDR_{532}$ varies significantly from 13 September to 9 October 2019, suggesting the air pollution contributed by different aerosol types. From 15 September to 18 September, and 19 September to 23 September, the $PLDR_{532}$ exceeds 0.25 above 1 km, and the value of $Bac_{532}$ ranges from 2 to 4 $Mm^{-1}$ $sr^{-1}$. Thus, there was a transmission of Asian dust in the lower free troposphere. However, the $PLDR_{532}$ below 1 km is between 0.02 and 0.2, and the maximum $Bac_{532}$ reaches 56 $Mm^{-1}$ $sr^{-1}$, indicating extremely heavy air pollution. Particularly, the $PLDR_{532}$ is less than 0.08 while $Bac_{532}$ is greater than 12 $Mm^{-1}$ $sr^{-1}$. Therefore, the upper Asian dust and lower urban haze mixed unevenly during the two consecutive air pollution processes of 15 September to 18 September and 19 September to 23 September, and the aerosol was stratified. The $PLDR_{532}$ from 24 September to 3 October ranges from 0.1 to 0.2, which is a typical mixed aerosol type (urban haze and Asian dust). The aerosol was stratified from 5 October to 8 October, the lower $PLDR_{532}$ is less than 0.1 (urban haze), while the maximum value of upper $PLDR_{532}$ reaches 0.36 (Asian dust).

The vertical distribution of air pollutants mass concentrations in the troposphere can better represent the evolution of air pollution and help us better understand the vertical distribution of atmospheric particulate matter. Applying the method in Section 3.1, the mass concentrations of natural dust and urban haze were obtained by combining the PRL and sun photometer observations. The overlap between the laser beam of PRL and its receiving system's file of view is less than 1 from surface to approximately 250 m. Thus, here we compare the particle mass concentrations at 250 m retrieved by PRL with the surface $PM_{10}$ and $PM_{2.5}$ to evaluate the method in Section 3.1. The $Bac_{532}$ at 250 m is positively correlated with the surface $PM_{2.5}$ mass concentrations, Pearson correlation R = 0.81, $p < 0.01$ (Figure 3). For mass concentrations, the Pearson correlation coefficient between urban haze mass concentrations and surface $PM_{2.5}$ is 0.74 ($p < 0.01$), and the Pearson correlation coefficient between total mass concentrations and surface $PM_{10}$ is 0.69 ($p < 0.01$). Due to the influence of the water vapor effect [43], it is difficult to retrieve completely consistent results in comparison, i.e., $Bac_{532}$ vs. $PM_{2.5}$. Moreover, the $PM_{2.5}$

and $PM_{10}$ were measured at the ground 15 km north of BMOC, while $Bac_{532}$ and $PLDR_{532}$ is the value at 250 m in BMOC. However, the significant positive correlation in Figure 3 shows that our method can estimate the vertical distribution of particulate matter mass concentrations of dust and urban haze in Beijing.

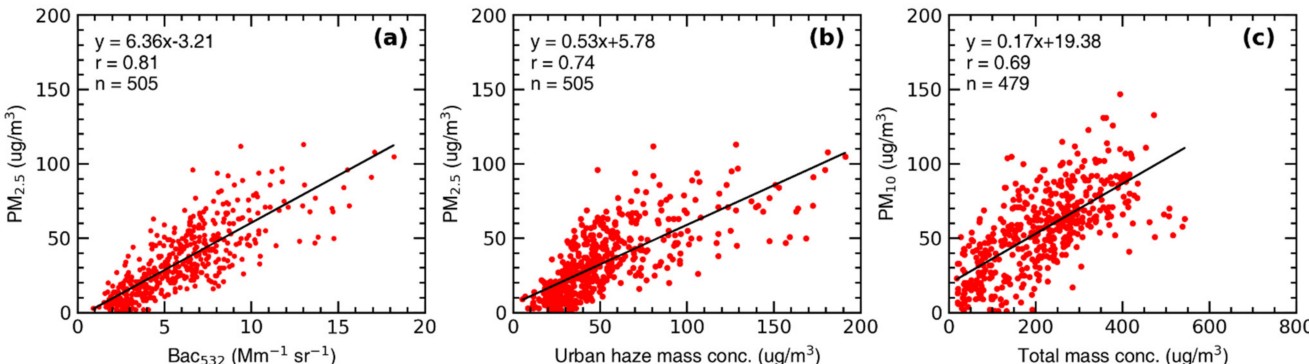

**Figure 3.** Scatter plots show the correlation between the (**a**) hourly average $Bac_{532}$ at 250 m and surface $PM_{2.5}$, (**b**) hourly average urban haze mass concentrations at 250 m and surface $PM_{2.5}$, and (**c**) hourly average total mass concentrations at 250 m and surface $PM_{10}$.

It is also found that the particle mass concentrations retrieved by PRL are greater than the surface observations, and the fitting slope is less than 1 (Figure 3). There may be two reasons. First of all, there is a cross-regional transmission of dust and urban haze at high altitudes, which makes the air pollution in the lower lidar layer is heavier than that on the ground, and it will be analyzed in detail in the following discussions. The second reason may be the different particle size response range. For example, the particle mass concentrations of urban haze particles obtained by PRL are not limited to a particle size of less than 2.5 μm, as long as the spherical particles with low $PLDR_{532}$ are counted into the mass concentrations of urban haze.

Among the four haze episodes, air pollution was contributed by natural dust and urban haze. The most serious and longest aerosol pollution was from 24 September to the evening of 3 October. The average surface mass concentrations of $PM_{10}$ and $PM_{2.5}$ were 71.3 μg/m$^3$ and 48.5 μg/m$^3$, respectively. It began to rain at 20:00 on 3 October, and the surface mass concentrations of $PM_{10}$ and $PM_{2.5}$ dropped rapidly below 10 μg/m$^3$. The PRL stopped observation during the rain, as shown in the white area in Figure 2. After the rain stopped, the $Bac_{532}$ retrieved by PRL was less than $2 \ Mm^{-1} \ sr^{-1}$, and the $PLDR_{532}$ was around 0.05. In addition, at 10:00 on 1 October 2019 (LT), a grand military parade was held at Tiananmen Square in Beijing to celebrate the 70th anniversary of the founding of the People's Republic of China. Beijing has implemented a series of emission controls, such as traffic control in urban areas, suspension of construction sites, and banning fireworks. With low local emissions, this provides a valuable opportunity to quantify the relative contribution of cross-regional transmission of anthropogenic and natural sources to air pollution in a megacity.

### 4.2. Characteristics of Dust and Urban Haze Particle Mass Concentrations during the 2019 National Day Military Parade

The haze episode from 24 September to 3 October was selected to investigate the vertical distribution of dust and urban haze mass concentrations in the lower troposphere. Applying the method in Section 3.1, the time–height distribution of dust and urban haze mass concentrations were shown in Figure 4.

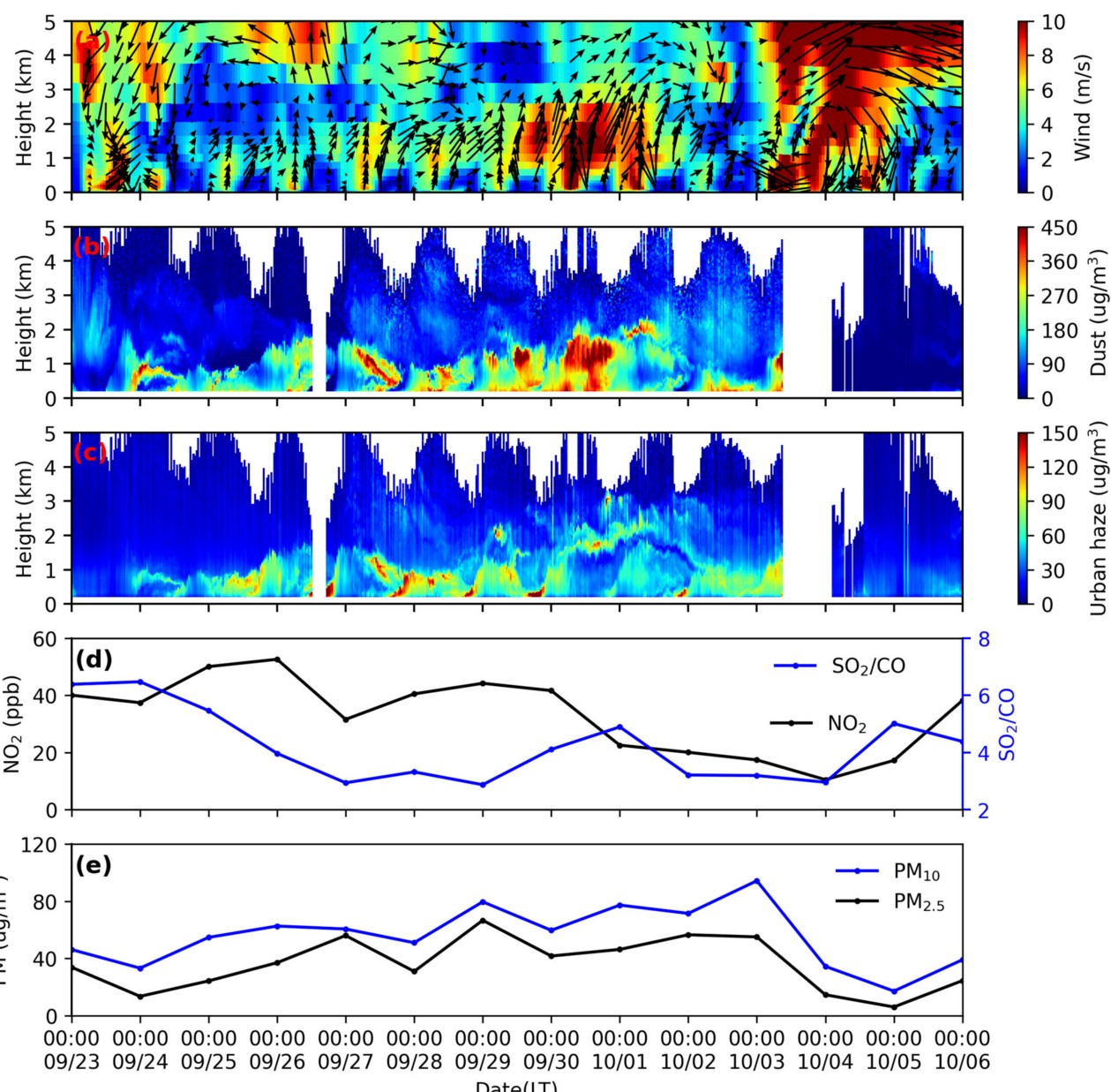

**Figure 4.** Air pollution in Beijing during the 2019 National Day military parade. The vertical structure of (**a**) horizontal winds simulated by WRF–Chem, (**b**) dust mass concentrations retrieved by PRL, and (**c**) urban haze mass concentrations retrieved by PRL. (**d**) Daily average $NO_2$ concentrations and $SO_2/CO$ ratios. (**e**) Daily average $PM_{2.5}$ and $PM_{10}$ mass concentrations. The black arrows in (**a**) represent the wind direction, and the downward arrow indicates the north wind.

Nature dust and urban haze are mainly concentrated below 3 km, and the maximum mass concentrations of dust and urban haze are 610 $\mu g/m^3$ and 221 $\mu g/m^3$, respectively. The vertical structure of dust and urban haze was similar from 23 to 29, September, indicating that natural dust and urban haze mixed well, while the mass concentrations of natural dust are about three times that of urban haze. From the afternoon of 29 September to 2 October, there was urban smoke plume transmission between 2 and 3 km, with a mass concentration of 80–120 $\mu g/m^3$. There was also the accumulation of urban haze in the lower lidar layer, the mass concentrations range from 40 to 70 $\mu g/m^3$, and the stratification of urban haze occurred. High loading dust aerosols extended from 0.25 to 2.0 km (200–500 $\mu g/m^3$) from 23 to 29 September, and there was also a transmission of natural dust between 2 and 5 km, the mass concentrations were less than 90 $\mu g/m^3$. From the afternoon of 1 to 2 October, dust was transported at 1–3 km and below 1 km, and dust aerosols also stratified. The stratification of dust and urban haze below 3 km was mainly caused by unstable meteorological conditions. The horizontal wind speed below 3 km increased significantly (exceeding 10 m/s) from 29 September to 2 October (Figure 4a). Strong southwesterly winds brought a mixture of dust and urban haze into Beijing, causing severe air pollution. The heavy haze episode lasted until 20:00 on 3 October, after which the air quality improved significantly due to rainfall, with $PM_{2.5}$ mass concentrations less than 10 $\mu g/m^3$.

We also show the surface $NO_2$ concentrations and $SO_2/CO$ ratio. Due to the low industrial concentration and large flow of motor vehicles in Beijing, Beijing has low $SO_2$ emissions and high CO emissions. Therefore, the $SO_2/CO$ ratio can reflect the contribution of local emissions to air pollution, and the lower the ratio, the greater the contribution of local emissions [28]. As can be seen in Figure 4 from 29 September to 1 October, with the increase of wind speed, the $SO_2/CO$ ratio gradually increased. In addition, the surface $NO_2$ concentrations dropped rapidly after 30 September, and the $PM_{10}$ and $PM_{2.5}$ mass concentrations gradually increased from 24 September to 3 October. The observation site is near the South Fifth Ring Road in Beijing, and the surrounding traffic volume is high. The decrease in surface $NO_2$ concentration may be due to traffic control and holidays. During the National Day military parade, workers and students have holidays, and factories are closed. The high $SO_2/CO$ ratio, increased $PM_{10}$ and $PM_{2.5}$ mass concentrations, as well as low $NO_2$ concentrations, indicating that during the National Day military parade, serious air pollution was mainly affected by regional transmission, and the contribution of local emissions was insignificant.

CALIPSO had two overpasses that passed over the North China Plain at 2:24 and 13:01 on 1 October, and the two overpasses were approximately 44.1 and 12.8 km away from the BMOC, respectively. Figure 5 shows the variation of 34–44°N aerosol vertical structure over the North China Plain measured by CALIPSO. There is a thick aerosol layer at 2–3 km over the whole North China Plain on 1 October, the extinction coefficient at 532 nm ranges from 0.3 to 0.5, and the $PLDR_{532}$ is less than 0.1. The VFM results show that the thick aerosol layer at 2–3 km is urban haze plumes (elevated smoke, black), and under the plumes are mainly a mixture of urban haze and dust (polluted dust, brown). The CALIPSO observations also confirmed that on 1 October, a mixture of urban haze and dust aerosols was transported from the ground to the upper air due to the development of the atmospheric boundary layer, while the urban haze plumes were transported at higher altitudes.

### 4.3. Quantify the Dust and Urban Haze Concentrations to Air Pollution during the 2019 National Day Military Parade

We further quantified the contribution of dust and urban haze cross-regional transmission to air pollution in Beijing. Figure 6 shows the daily average dust and urban haze mass concentrations profiles retrieved by PRL. Figure 7 shows the night-time mean $LR_{355}$ profiles, as well as daily average profiles of the $PLDR_{532}$. The mass concentrations of urban haze increased continuously from 30 μg/m$^3$ to 85 μg/m$^3$ from 24 to 29 September, and the dust mass concentrations increased from 100 μg/m$^3$ to 230 μg/m$^3$ from 24 to 29 September. With the strengthened wind speed from 28 September to 1 October, the aerosols were transported from the ground to the upper air, and the height of the dust layer was rising, from 1 km on 28 September to 2.5 km on 1 October. The dust mass concentrations in the lower lidar layer were between 200 and 320 μg/m$^3$ from 28 September to 1 October. The maximum dust mass concentrations in the lower lidar layer appeared on 30 September, reaching 320 μg/m$^3$. Similarly, the urban haze layer also keeps rising from 28 September to 1 October, from 1 km on 28 September to 3.3 km on 1 October, and the mass concentrations of urban haze in the lower lidar layer are between 50 and 80 μg/m$^3$.

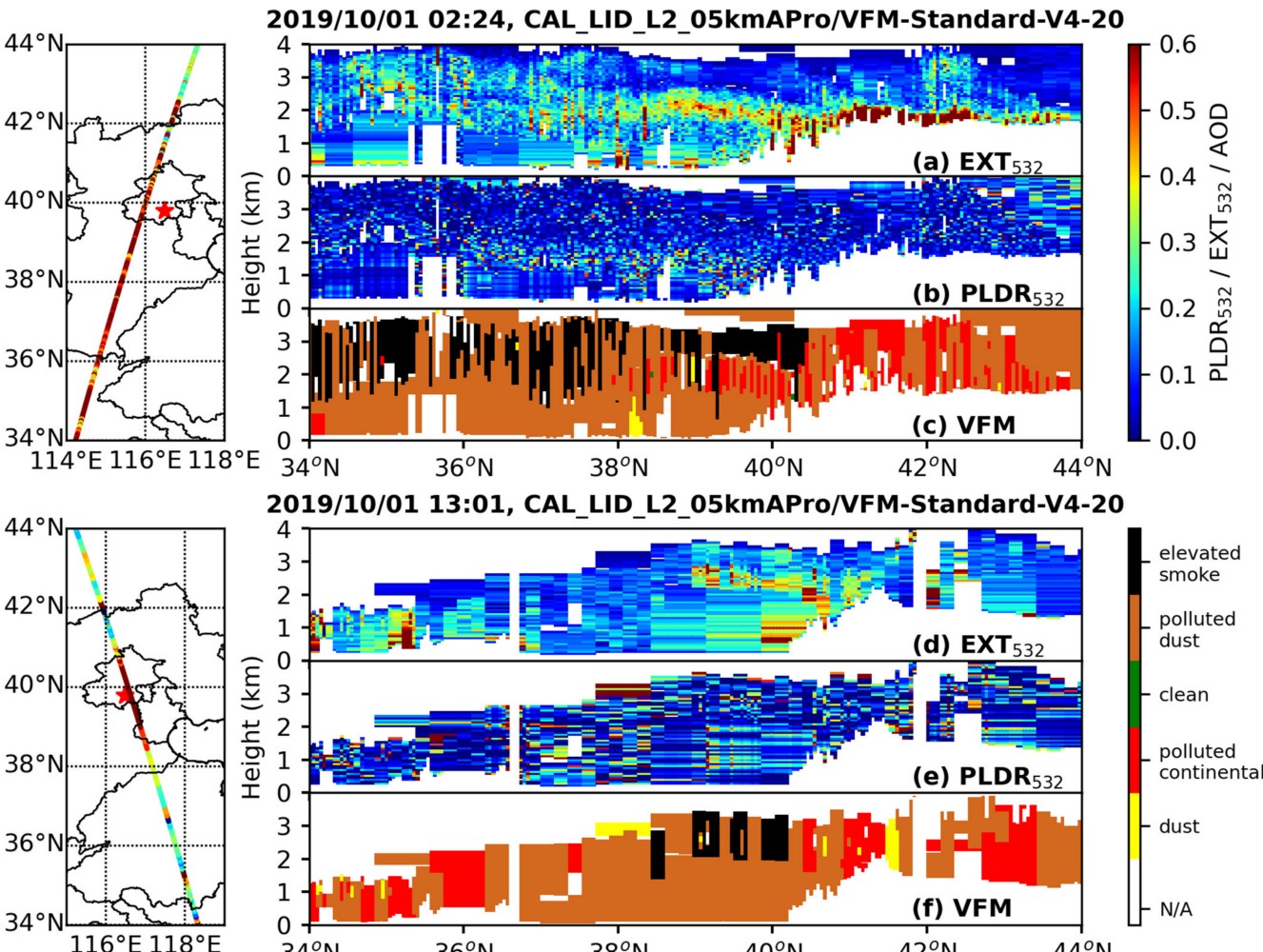

**Figure 5.** The CALIPSO measurements in North China on 1 October 2019. Latitude–height plots of the (**a**) extinction coefficient at 532 nm, (**b**) $PLDR_{532}$, and (**c**) VFM at 2:24 LT. Latitude–height plots of the (**d**) extinction coefficient at 532 nm, (**e**) $PLDR_{532}$, and (**f**) VFM at 13:01 LT. The left two panels are the CALIPSO ground tracks color-coded by AOD. The red star on the map indicates BMOC.

On the day of the military parade (1 October), the $PLDR_{532}$ and $LR_{355}$ below 1.5 km were 0.14–0.19 and 40–45 sr, respectively, and the $PLDR_{532}$ and $LR_{355}$ above 2 km were

0.1–0.12 and 50–70 sr, respectively (Figure 7). High concentrations of dust (150–250 µg/m$^3$) extended from the ground to 2.0 km, while the urban haze transmitted at a higher altitude, from the ground to 3.0 km, and the mass concentrations range from 30 to 60 µg/m$^3$. Natural dust is the main air pollutant on 1 October, with a relative contribution of more than 72% below 2 km (Figure 8). The PLDR$_{532}$ and LR$_{355}$ from 24 September to 3 October are 0.13 ± 0.05 and 52 ± 9 sr, respectively. It is the typical value of a mixture of dust and urban haze [31]. The average relative contributions of urban haze mass concentrations in total mass concentrations from 24 September to 3 October is less than 26%. Thus, the cross-regional transmission of natural dust dominated air pollution during the 2019 National Day military parade, with a relative contribution of approximately 78%.

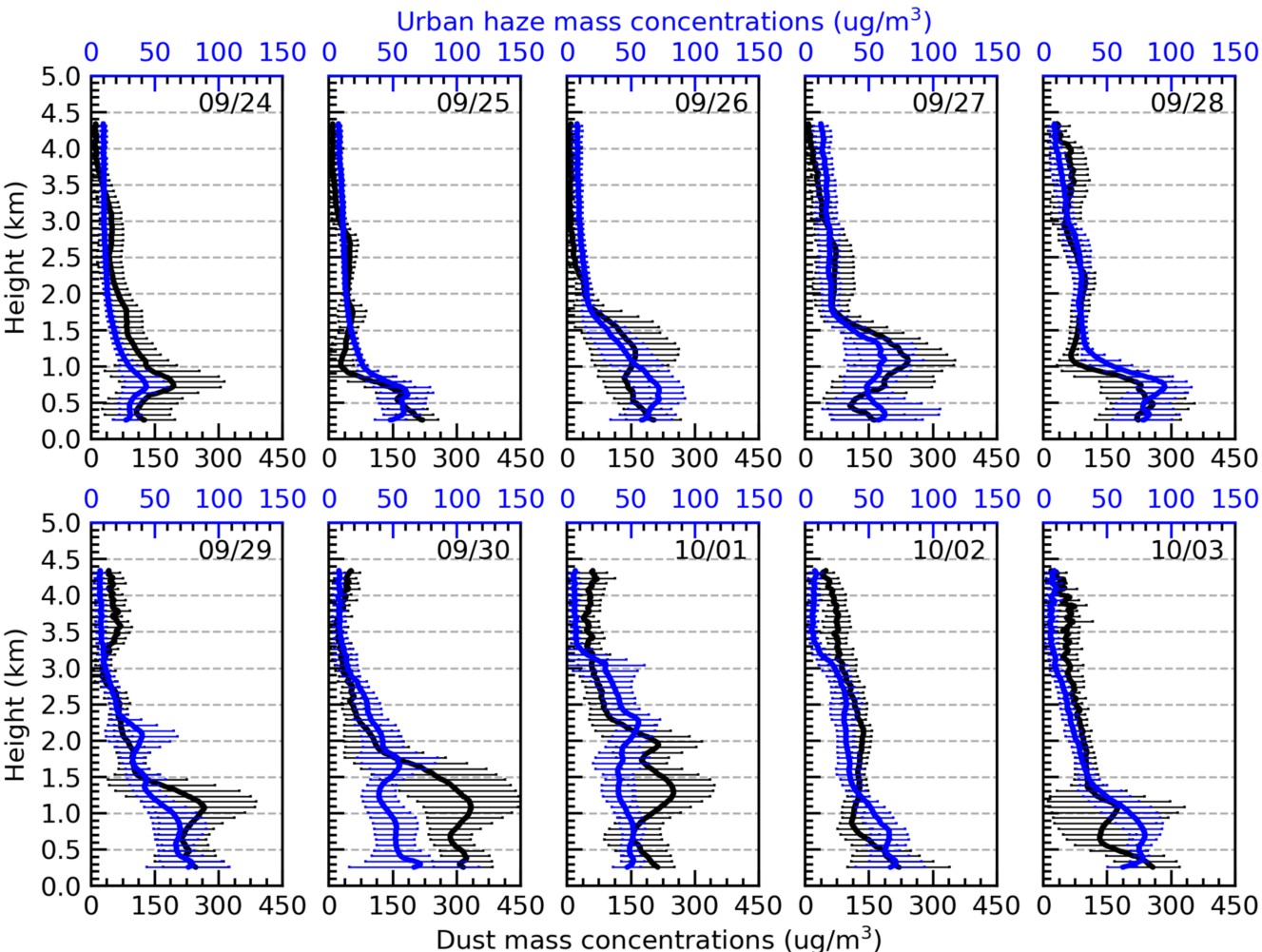

**Figure 6.** Height profiles of the daily average dust mass concentrations (**black**) and urban haze mass concentrations (**blue**). The envelopes in dust and urban haze mass concentrations represent the one standard deviation at each height. The date is displayed at the top right of each panel.

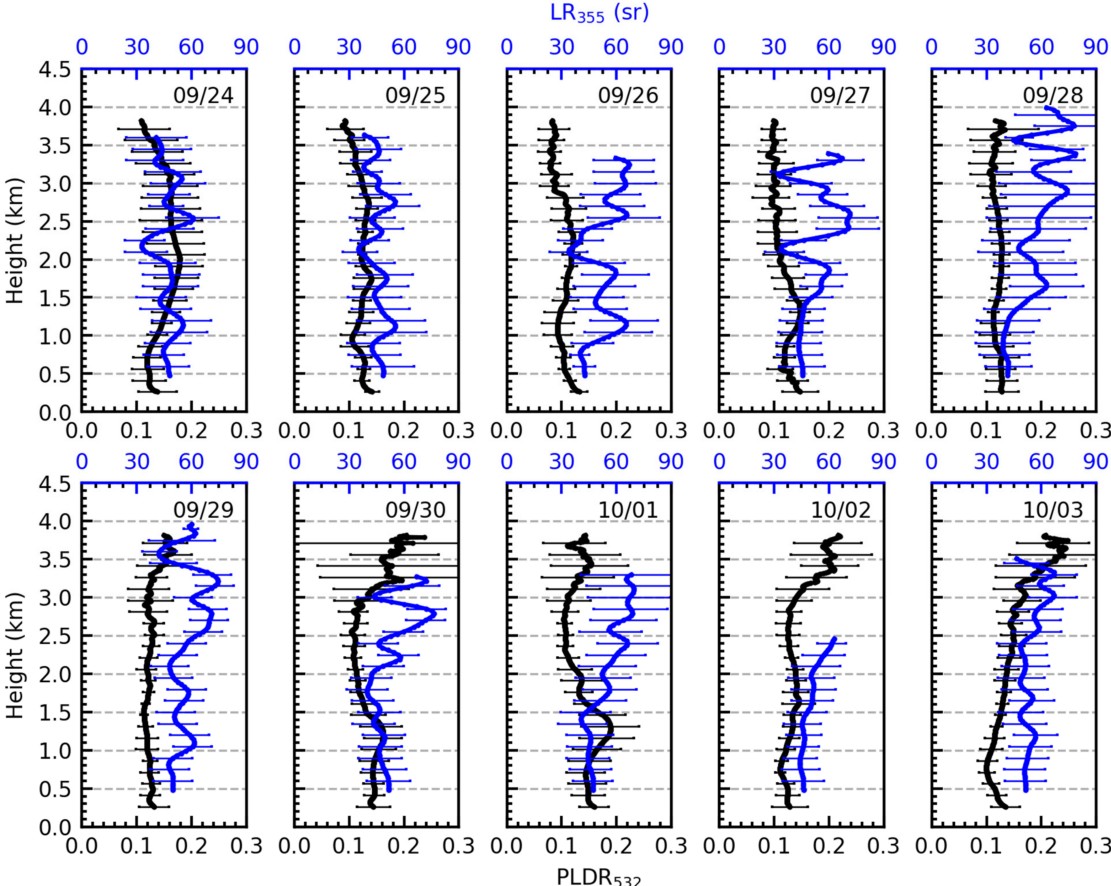

**Figure 7.** Height profiles of the daily average $PLDR_{532}$ (**black**) and night-time average $LR_{355}$ (**blue**). The envelopes in $PLDR_{532}$ represent the one standard deviation at each height, and the envelopes in $LR_{355}$ indicate the errors at each height. The date is displayed at the top right of each panel.

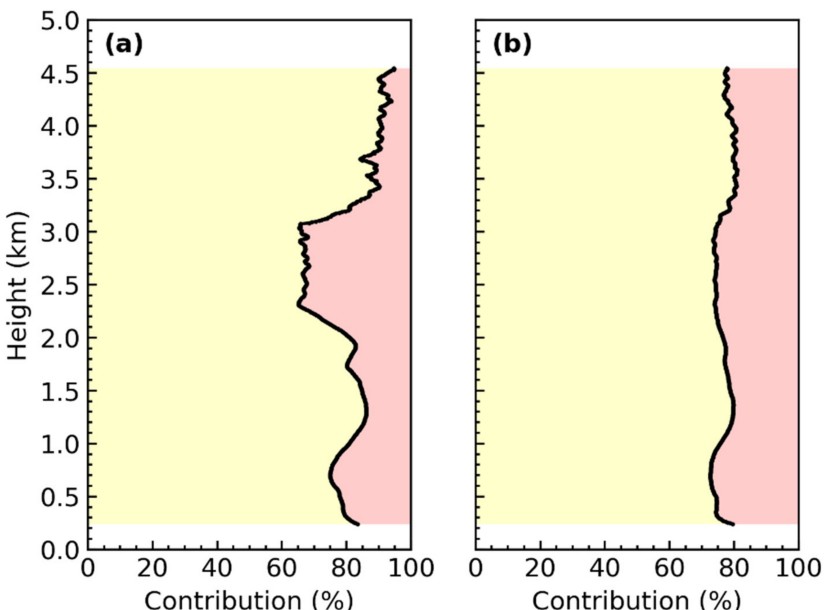

**Figure 8.** The average relative contributions of dust (yellow shade) and urban haze (red shade) mass concentrations in total mass concentrations (**a**) on 1 October, and (**b**) from 24 September to 3 October.

## 5. Discussion

It would be interesting to assess how anthropogenic emissions and natural dust can interfere with regional transport and affect air quality in Beijing. We investigated the potential sources of air pollution during the 2019 National Day military parade by MODIS measurements. Figure 9 presents the spatial distribution of the daily AOD (MODIS MCD19A2 product) from 24 September to 3 October 2019. Atmospheric pollutants began to accumulate in the south of Beijing (Southern Hebei, Shandong) on 26 September. Strong surface wind (almost 10 m/s) caused the increased low-level air pollutants to gradually move to Beijing. Based on the daily AOD from 24 September to 3 October 2019. We further investigated the spatial correlation between the AOD at BMOC and the AOD at the investigated pixel. Figure 10 shows the spatial distribution of AOD correlation coefficients with 0-day, 1-day, and 2-day lag coefficients. The higher correlation means the closer relationship between the BMOC and the investigated pixel, which indicates the stronger the influence of regional transportation, and vice versa. Areas with an R-value greater than 0.5 can be considered as potential air pollution sources that are more likely to cause disturbances to Beijing's air quality.

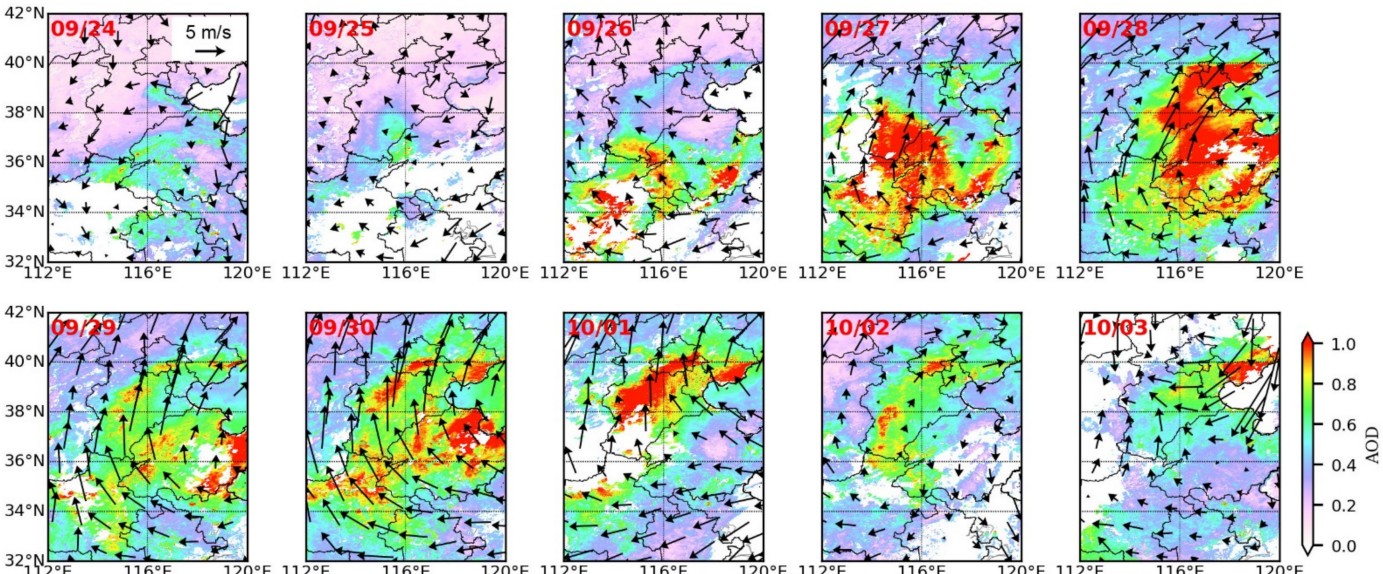

**Figure 9.** Spatial distribution of the daily AOD from 24 September to 3 October 2019 retrieved from the Moderate Resolution Imaging Spectrometer (MODIS). The black arrows overlaid on the map represent the surface wind speed and direction. The date is displayed at the top left of each panel.

A strong spatial correlation was found in Hebei and northern Shanxi (above 38°N) when the lag days were 0, with an R-value of 0.8–1.0. With the increase in lag days, the spatial correlation above 38°N decreased considerably, and the spatial correlation in Shanxi, Henan, Shandong, and Shaanxi gradually increased, the correlation coefficient ranges from 0.5 to 0.9. In addition to anthropogenic emissions in the North China plain, northwestern China (above 40°N), including the Mongolia region, northwestern Gansu, and northern Xinjiang also showed a significant correlation of 0.5–0.8. Overall, potential anthropogenic emission sources affecting Beijing's air quality include most areas of the North China plain and Guanzhong plain, and the potential natural dust that affects Beijing's air quality is mainly concentrated in the Mongolia region and Xinjiang.

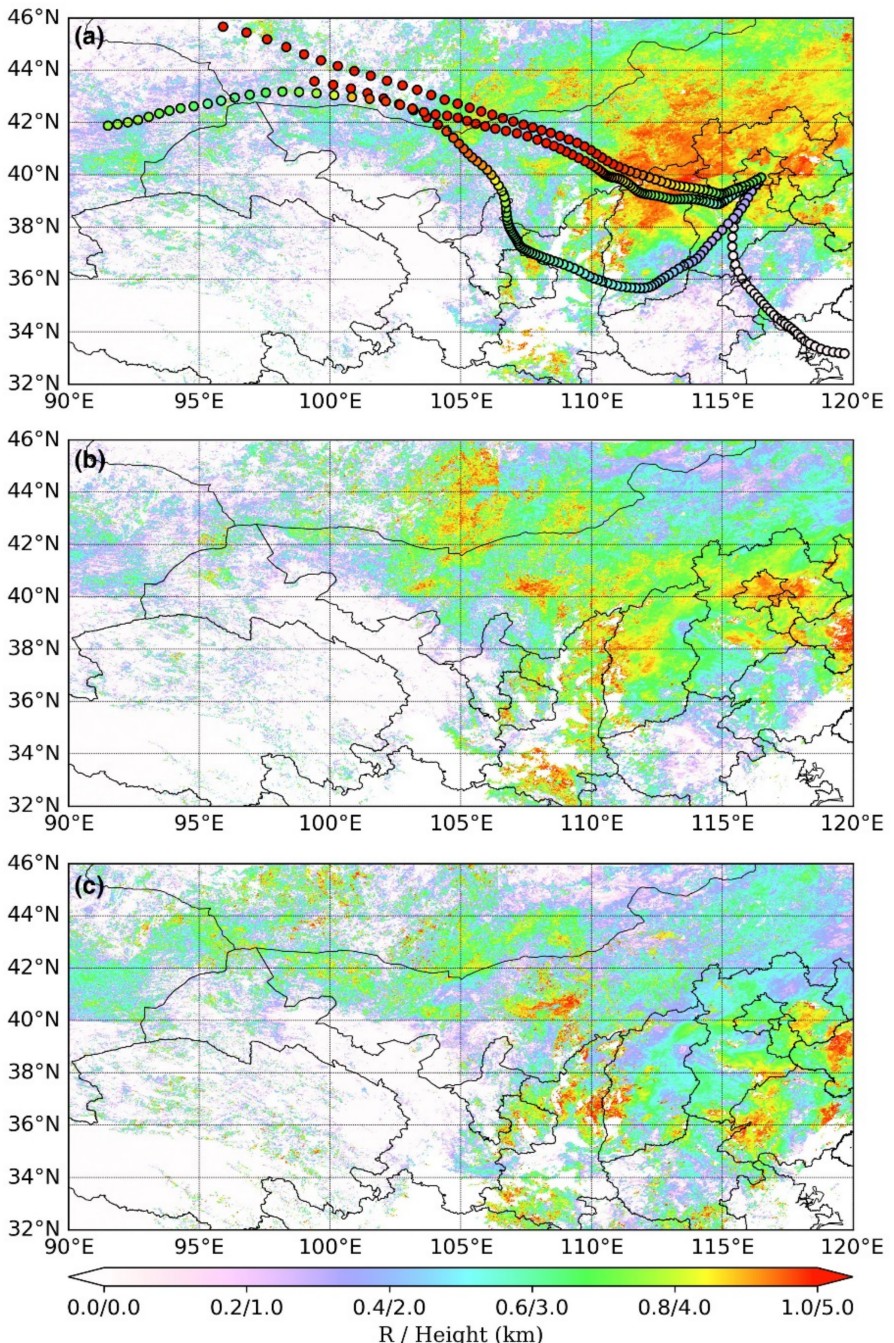

**Figure 10.** Spatial distribution of the correlation coefficients between observation site and each grid's daily AOD at lag (**a**) 0 day, (**b**) 1 day, and (**c**) 2 day from 24 September to 3 October, 2019. Seventy-two hour backward trajectories arriving at the observation site at 10:00 LT on 1 October 2019 at 500 m, 1500 m, 2500 m, and 3500 m overlaid on (**a**).

The HYSPLIT model was used to analyze the 72 h backward trajectories at 10:00 LT on 1 October 2019 (LT) at 500 m, 1500 m, 2500 m, and 3500 m. The air mass at 500 m comes from southeast China, one of the most serious regions of anthropogenic emission [1,2]. Based on MODIS observations, the AOD in North China plain (especially Shandong and Hebei province) is over 1.0. The air masses at 1500 m were from the source regions of dust (Taklimakan desert and Gobi desert), passing through Shaanxi, Hebei province, causing the dust to mix with local anthropogenic pollutants. High-altitude air masses at 2500 m and 3500 m are from the Mongolia regions. After long-range transportation, the height of the dust layer in Beijing drops to 1–2 km. Thus, the low-level air pollutants are contributed by

south transmission (urban haze), while the upper air pollutants come from the Mongolia region (Asian dust).

## 6. Conclusions

Based on ground-based and satellite remote sensing, our research quantifies the contribution of cross-regional transported natural dust and anthropogenic pollutants to air quality in Beijing by taking air pollution during the 2019 National Day military parade as an example and analyzes the potential sources of different aerosol types. The main research results are summarized as follows.

1.  There is a good correlation between the dust and urban haze mass concentrations retrieved by PRL and surface $PM_{2.5}$ and $PM_{10}$. It shows that PRL can be used to investigate the fine structure of particulate matter profiles, and to quantify the contribution of anthropogenic and natural sources to air pollution, which is difficult to achieve by ground or satellite observations.
2.  During the 2019 National Day military parade, the contributions of local emissions to air pollution were insignificant, mainly affected by regional transport, including urban haze in North China plain and dust aerosol in northwestern China. The dust and urban haze are more evenly mixed after arriving in Beijing. Dust aerosols dominate air pollution, and their relative contribution to particulate matter mass concentrations exceeds 74%. In addition, Wet deposition can significantly improve air quality.
3.  Through spatial correlation analysis, we found that the potential emission sources that affect Beijing's air quality include North China Plain and Guanzhong Plain, mainly concentrated in Hebei, Tianjin, Shandong, and Shanxi. Our results indicate that controlling anthropogenic emissions over regional scales is crucial and effective to improve Beijing's air quality. More importantly, consider the effects of natural dust in northwest China, it can lead to heavy air pollution in Beijing in the short term.

Our results clarified the sources of air pollution in Beijing during the 2019 National Day military parade and quantified the natural dust and anthropogenic contributions, which can provide a scientific basis for the control and management of atmospheric emissions in northern China, as well as for the design and implementation of regional coordinated emission reduction strategies. Although the results have given the contributions of anthropogenic and natural sources to air pollution in Beijing, these are still rough estimates for a short time. The detailed contributions of anthropogenic and natural sources in northern China still need further investigation, especially long-term measurements, although it is beyond the scope of the current analysis.

**Supplementary Materials:** The following are available online at https://www.mdpi.com/article/10.3390/rs13163326/s1, Figure S1: Wind speed of u components, Figure S2: Wind speed of v components, Figure S3: Relative humidity.

**Author Contributions:** Conceptualization, C.L., C.X. and Z.W.; data curation, Z.W., T.L., Y.Z. and Y.D.; formal analysis, Z.W.; funding acquisition, C.L. and Q.H.; investigation, Z.W., C.L., C.X. and Y.D.; methodology, Z.W., C.L., C.X. and Y.D.; project administration, Z.W., Q.H., C.X. and Y.D. All authors have read and agreed to the published version of the manuscript.

**Funding:** This research was supported by grants from the National Key Research and Development Program of China (No. 2017YFC0210002, 2018YFC0213201, and 2018YFC0213104), the National Natural Science Foundation of China (No. 41722501, 51778596, and 41977184), Anhui Science and Technology Major Project (No. 18030801111), the Strategic Priority Research Program of the Chinese Academy of Sciences (No. XDA23020301), the Major Projects of High-Resolution Earth Observation Systems of National Science and Technology (05-Y30B01-9001-19/20-3), the Youth Innovation Promotion Association of CAS (2021443), the Young Talent Project of the Center for Excellence in Regional Atmospheric Environment, CAS (CERAE202004).

**Data Availability Statement:** All code/data needed to evaluate the conclusions in the paper are present in the paper. Additional code/data related to this paper may be requested from the authors.

**Acknowledgments:** The authors acknowledge the National Oceanic and Atmospheric Administration (NOAA) Air Resources Laboratory (ARL) for the provision of the HYSPLIT transport and dispersion model used in this publication. We thank NASA Langley Research Center Atmospheric Sciences Data Center for providing the CALIPSO data. We'd like to sincerely thank the MODIS data processing team members for making the data available.

**Conflicts of Interest:** The authors declare no conflict of interest.

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
