# Peer review of "Profiling of Dust and Urban Haze Mass Concentrations during the 2019 National Day Parade in Beijing by Polarization Raman Lidar"

_remotesensing, doi:10.3390/rs13163326_

Round 1

Reviewer 1 Report

This study performed an observation at Beijing during the 2019 National Day parade and quantified the contributions of cross-regional transported natural dust and urban haze mass concentrations to Beijing’s air quality. The results can provide a scientific basis for the control and management of atmospheric emissions in northern China, as well as for the design and implementation of regional coordinated emission reduction strategies. From this aspect, the work presented here is a useful supplement for scientific community, public and government community. This paper is well-written, well-structured and is a welcome addition to the fields and describes some nice results. I believe it can be accept after minor revisions as detailed below:

  1. Some important information about the methodology is missing in the paper. Needs add a part in section 3 to describe the contribution calculation method of dust and urban haze mass concentrations.
  2. I do not ask for a refit with different periods, but the comparisons between your studies and other related studies with different period (when hosting major events or spring festival in Beijing) are high recommended.
  3. It is useful to describe the performance of the WRF-Chem simulated results in this study. At least, you should validate the model performance with the precision, for readerships.
  4. How about the water vapor content and its effect on dust and urban haze mass concentrations, which need to be discussed here.
  5. Line 4: lidar-->> Lidar

Reviewer 2 Report

The study is complete and the work carried out with seriousness. The use of the PLR ​​is supplemented by other sources of information and the results are compared to several approaches:
- two satellites (MODIS and CALIPS)
- Two dispersion models (Hysplit / retrotrajectory - WRF-CHEM / CTM
- ground measurement network
Question 1: Dust, haze and ground PM10 Mesurement
The separation between the dust and the haze is not very clear but seems to work. We are however surprised that the concentrations obtained by the analysis of the LIDAR signals greatly exceed 150 µg / m3 in the low layers both for the dusts and for the haze whereas the ground stations (BMOC) supposed to measure the totality only exceed rarely the 150 µg / m3?
Question 2 use of WRF-CHEM output
The only outputs of WRF-CHEM shown in this paper are the winds (fig 4). A WRF-CHEM sensitivity study could also have confirmed the relative importance of local and distant pollution?

Reviewer 3 Report

The current study is based on a case study on air quality characterization during a national day parade event in Beijing during Fall of 2019. For this study, the authors have used about one month of data collected using various ground based and satellite instruments. Following are few comments and suggestions to consider:

Page 6 (lines 198 & 206): What does 0:01 signify? Is it local time or GMT?

What was the timing during which the parade occurred on Oct. 1st?
It would be nice if the authors give some statistics on ground level pollution on the day of the parade and compare it with the previous day and next day and discuss this in the context of the influence of the parade day (more/less traffic) on the observations.

Lines 228 - 230: The two points (and threshold > 5) to calculate the Pearson correlation corresponds to how much data averaging in terms of timing (approximately how much minutes/hours)?

Lines 456-457:  Consider replacing "wet deposits" by "Wet deposition". 
However, wet deposition is not explicitly discussed in the results section. It has only been mentioned in the conclusions.
